# LAST ONE STANDING: A COMPARATIVE ANALYSIS OF SECURITY AND PRIVACY OF SOFT PROMPT TUNING, LORA, AND IN-CONTEXT LEARNING

## ABSTRACT

Large Language Models (LLMs) are powerful tools for natural language processing, enabling novel applications and user experiences. However, to achieve optimal performance, LLMs often require adaptation with private data, which poses privacy and security challenges. Several techniques have been proposed to adapt LLMs with private data, such as Low-Rank Adaptation (LoRA), Soft Prompt Tuning (SPT), and In-Context Learning (ICL), but their comparative privacy and security properties have not been systematically investigated. In this work, we fill this gap by evaluating the robustness of LoRA, SPT, and ICL against three types of well-established attacks: membership inference, which exposes data leakage (privacy); backdoor, which injects malicious behavior (security); and model stealing, which can violate intellectual property (privacy and security). Our results show that there is no silver bullet for privacy and security in LLM adaptation and each technique has different strengths and weaknesses.

## 1 INTRODUCTION

In recent years, Large Language Models (LLMs) have become integral to a plethora of products. Their efficacy is further underscored by their ability to adapt to customized –possibly private or personal– domains. Among the existing adaptation techniques, three have been particularly salient. First is Low-Rank Adaptation (LoRA) (Hu et al., 2022), wherein rank decomposition matrices are inserted into the target model enabling its recalibration to accommodate new datasets. Second, the Soft Prompt Tuning (SPT) (Lester et al., 2021) method, which optimizes prompt tokens with respect to the new dataset, and then prepends it to the inputs' embeddings. Finally, In-Context Learning (ICL) (Zhao et al., 2021) where selected samples from the new dataset are placed directly into the input, serving as illustrative exemplars of the new dataset task/distribution.

Despite some studies exploring the variations in utility among various adaptation techniques, a noticeable gap exists in the comprehensive comparison of their security and privacy properties. This paper takes a step to fill this gap, offering a three-fold assessment that encompasses both privacy and security aspects. In terms of privacy, our evaluation centers on the resilience of these techniques against one of the most well-established privacy concerns: membership inference attacks (MIAs).

On the security front, we study the robustness of these techniques against two severe security threats. The first entails model stealing, wherein we evaluate the likelihood of an adversary successfully replicating the adapted model. The second revolves around backdoor attacks, where an adversary seeks to poison the dataset with the intention of embedding a stealthy backdoor into the model. Such a backdoor, if exploited, would empower the adversary to control the model's output, e.g., outputting a specific response or label, by introducing a predefined trigger.

We conduct an in-depth evaluation across three different LLM architectures: GPT2 (Radford et al., 2019), GPT2-XL(Radford et al., 2019), and LLaMA (Touvron et al., 2023), using four recognized NLP benchmark datasets: DBPedia (Zhang et al., 2015), AGNews (Zhang et al., 2015), TREC (Li and Roth, 2002), and SST-2 (Wang et al., 2019). Figure 1 provides an abstract comparison of ICL, LoRA, and SPT with respect to membership inference attacks, model stealing, and backdoor threats. The figure highlights the lack of a single superior technique resilient against all privacy and security threats. For example, while ICL shows strong resistance to backdoor attacks, it is more vulnerable to

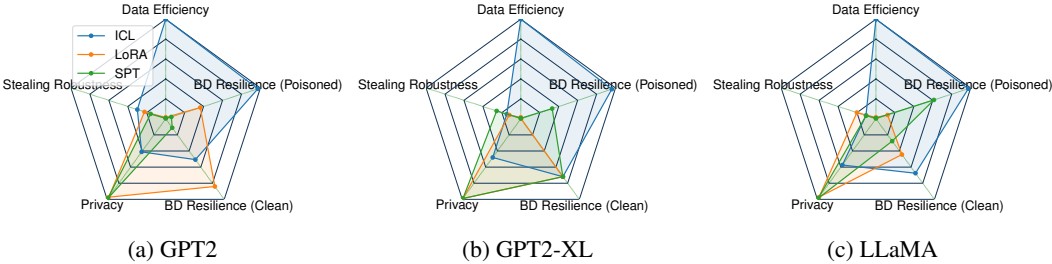

Figure 1: Comparative overview of ICL, LoRA, and SPT: Evaluating Privacy (resilience against membership inference attacks), Model Stealing Robustness (difficulty of unauthorized model replication), Data Efficiency (based on required training dataset size), and Backdoor Resilience with both Poisoned (backdoored/triggered data avoidance) and Clean (accurate label prediction) data scenarios. Larger values indicate better performance.

membership inference attacks. Therefore, choosing the appropriate technique heavily relies on the specific scenario at hand.

To the best of our knowledge, our detailed analysis is the first to extend some of the most prevalent attacks against machine learning models, such as the model stealing attack, into the domain of LLM with adaptation techniques. Furthermore, we believe it contributes valuable insights to the ongoing discourse on LLM adaptation techniques, offering a comprehensive view of their strengths and vulnerabilities. As the landscape of language models continues to evolve, our work provides a foundation for refining and advancing strategies that balance usability and privacy/security considerations in real-world applications.

## 2 RELATED WORK

**Training-efficient Adaptation Methods:** Training Large Language Models (LLMs) for customized domains presents significant challenges due to their extensive parameter sizes, necessitating considerable computational resources. To address these challenges, innovative, computationally-efficient methods have been developed. Low-Rank Adaptation (LoRA) (Hu et al., 2022) introduces rank-decomposition weight matrices, referred to as "update matrices", into the existing model parameters. The primary focus of training is shifted to these update matrices, enhancing training speed while simultaneously significantly decreasing computational and memory demands. Soft Prompt Tuning (SPT) (Lester et al., 2021) takes a different approach by adding a series of prompt tokens to the input. During training, SPT only updates the gradients of these prompt token embeddings, while keeping the pretrained model's core parameters frozen, making it computationally efficient. In-Context Learning (ICL) (Zhao et al., 2021) conditions the model directly on supplied demonstrations (which are samples that are introduced in the input to guide the model), thus avoiding parameter updates altogether. While these techniques are computationally advantageous, our analysis indicates potential vulnerabilities in terms of privacy and security.

**Attacks against LLMs:** Language models are vulnerable to a range of attacks, including membership inference (Mireshghallah et al., 2022; Hisamoto et al., 2020), reconstruction (Carlini et al., 2021), and backdoor (Chen et al., 2021; 2022) attacks. While much of the previous research has focused on the vulnerabilities of pretrained or fully fine-tuned models, we study the different efficient adaptation techniques, specifically ICL, LoRA, and SPT. We aim to assess their relative strengths and weaknesses in terms of various privacy and security properties. Although there are recent concurrent studies, like Kandpal et al. (2023), that investigate backdooring in-context learning, and others such as Duan et al. (2023a) that compare the information leakages (using membership inference) in fine-tuned models and in-context learning, our approach provides a more comprehensive comparison that encompasses additional training paradigms and datasets. Moreover, we extend the scope of comparison beyond privacy to include different security properties of the ICL, LoRA, and SPT techniques.

## 3 MEMBERSHIP INFERENCE

We begin by assessing the privacy attributes of the three adaptation techniques. To this end, we employ the membership inference attack (MIA), a recognized privacy attack against LLMs. Fundamentally, MIA aims to determine the likelihood of a given input being part of the training or fine-tuning dataset of a target model. In this work, the data used for training or fine-tuning corresponds to the datasets leveraged by the adaptation techniques, such as the demonstrations for ICL or the fine-tuning datasets for LoRA and SPT.

### 3.1 THREAT MODEL

We adopt the most conservative threat model, where the adversary is limited to black-box access to the target model. This scenario aligns with common deployment settings for LLMs, where the user merely obtains the label –specifically, the predicted words– along with their associated probabilities.

### 3.2 METHODOLOGY

We adopt the widely-used loss-based membership inference attack (Yeom et al., 2018), wherein we compute the loss for every target input. Notably, member samples often exhibit lower loss values when compared to non-member samples, as depicted in the appendix (Figure 12). This observation serves as the basis for our membership determination. To quantitatively evaluate the results, we adhere to the methodology outlined in the state-of-the-art MIA work (Carlini et al., 2022) that plots the true positive rate (TPR) vs. false positive rate (FPR) to measure the data leakage using a logarithmic scale. This representation provides an in-depth evaluation of data leakage, emphasizing MIA performance in the low FPR area, which better reflects the worst-case privacy vulnerabilities of language models.

In evaluating the privacy implications of the three distinct adaptation techniques—LoRA, SPT, and ICL—we strive to ensure a meticulous and fair comparison. Firstly, we first measure the utility of the ICL, recognizing its inherent constraint whereby the fixed input context length of target models limits the inclusion of demonstrations. Subsequently, we calibrate the hyperparameters of LoRA and SPT to align their performance with that of ICL, concrete model performance can be found in Appendix A. Following the training of these models, we employ membership inference attacks to assess their privacy attributes and draw comparative insights across the trio. Our assessment spans a variety of scenarios, integrating different datasets and target models, to thoroughly probe the privacy of ICL, LoRA, and SPT.

### 3.3 EVALUATION SETTINGS

We now outline our experimental setup for evaluating MIA against the adaptation techniques LoRA, SPT, and ICL. We use four well-established downstream text classification tasks, each featuring a different label count. These benchmarks, commonly used in adaptation methods evaluation, especially for In-Context Learning (ICL), include DBPedia (Zhang et al., 2015) (14 class), AGNews (Zhang et al., 2015) (4 class), TREC (Li and Roth, 2002) (6 class), and SST-2 (Wang et al., 2019) (2 class). Furthermore, we span our evaluation across three distinct language models: GPT2 (124M parameters) to GPT2-XL (1.5B parameters) and LLaMA (7B parameters).

To achieve comparable performance for the different adaptation techniques, we train the model with a varying number of samples. For example, with DBPedia, we use 800 (SPT) and 300 (LoRA) samples to fine-tune the model, where the number of demonstrations used for ICL is set to 4, detailed hyperparameter setting can be found in Appendix A. For ICL, we follow the prompt design by Zhao et al. (2021), which yields a good performance, examples can be found in the appendix (Table 1).

Following membership inference attack works (Shokri et al., 2017; Salem et al., 2019), we sample members and non-members as disjoint subsets from the same distribution. For both LoRA and SPT, we maintain an equivalent count for members and non-members. In the case of ICL, we follow previous works (Duan et al., 2023a) and consider more non-members (300) than members due to the constraint on the number of inputs in the prompt. To account for the inherent randomness, we conducted experiments 10 times for LoRA and SPT, and 300 times for ICL (due to its increased sensitivity of the examples used).

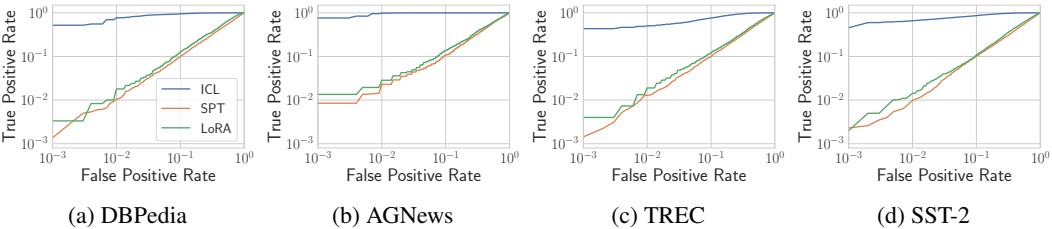

(a) DBPedia       (b) AGNews       (c) TREC       (d) SST-2

Figure 2: Membership inference attack performance using GPT2-XL across various datasets.

### 3.4 RESULTS

In Figure 2, we present the MIA performance across all four datasets using GPT2-XL as the target model. The figure clearly demonstrates that both Low-Rank Adaptation (LoRA) and Soft Prompt Tuning (SPT) have strong resistance to membership inference attacks, compared to ICL. Specifically, at a False Positive Rate (FPR) of $1 \times 10^{-2}$, both LoRA and SPT's performances align closely with random guessing. Quantitatively, LoRA and SPT achieve True Positive Rates (TPR) of $0.010 \pm 0.007$ and $0.011 \pm 0.004$, respectively. Conversely, In-Context Learning (ICL) exhibits significant susceptibility to membership inference attacks. For instance, when evaluated on the DBPedia dataset, ICL achieves a TPR of $0.520 \pm 0.237$ at the aforementioned FPR—a figure that is $52.0\times$ and $47.3\times$ greater than what LoRA and SPT respectively achieve.

We observe a similar pattern in the MIA performance across various datasets and models, as illustrated in Figure 2 and Figure 3. This can be attributed to the substantial differences in training data volume between ICL and the likes of LoRA and SPT. Specifically, ICL necessitates far fewer samples, often orders of magnitude less than what is required for SPT or LoRA. This observation aligns with previous membership inference studies which have highlighted that reduced training datasets tend to amplify the MIA success rates(Salem et al., 2019; Liu et al., 2022).

To further investigate the influence of training sample sizes on ICL, we assess the MIA attack using different sample counts, such as 4 and 8 demonstrations. The results, presented in Figure 4, confirm that as we increase the number of demonstrations, the susceptibility to MIA decreases. However, it is essential to highlight that given the model's limited context, there is a constraint on the maximum number of inputs that can be inserted. Consequently, we believe that MIA will consistently present a significant concern for ICL unless countered with an appropriate defense.

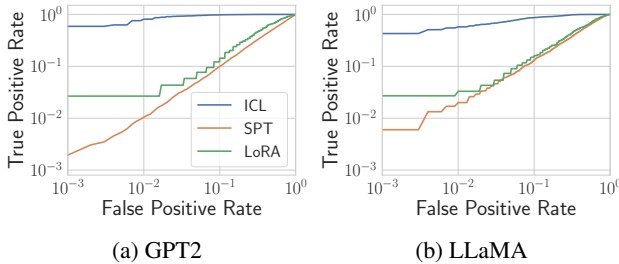

(a) GPT2       (b) LLaMA

Figure 3: Membership inference attack performance on GPT2 and LLaMA with the DBPedia dataset.

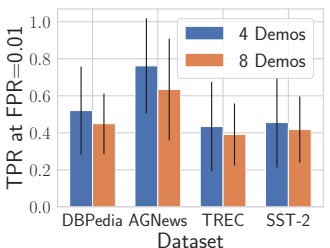

Figure 4: Membership inference attack with different number of demonstrations for ICL.

## 4 MODEL STEALING

Next, we examine the resilience of ICL, LoRA, and SPT against model stealing threats. In these scenarios, adversaries seek to illegally replicate the functional capabilities of the target LLM. It is important to recognize that organizations and individuals invest significant resources, including valuable data and computational power, in the development of optimal models. Therefore, the prospect of an unauthorized replication of these models is a substantial and pressing concern.

### 4.1 THREAT MODEL

We adopt the most strict settings following the same threat model as MIA (Section 3.1), where only the label and its probability are given. For this attack, our focus is solely on the label, making it applicable even to black-box models that do not disclose probabilities. However, we assume the adversary knows the base model, e.g., GPT2 or LLaMA, used in the target model. We believe that this assumption is reasonable, considering the unique performance characteristics demonstrated by various base LLMs.

### 4.2 METHODOLOGY

To steal the target model we follow previous works (Tramèr et al., 2016) and query the target model with a probing dataset. We explore two distinct strategies to construct this dataset. Initially, we assume the adversary has access to samples from the same distribution as the fine-tuning data. As an alternative, we utilize another LLM, specifically GPT-3.5-Turbo, to generate the probing dataset. This involves using the following prompt to generate the data *"Create a python list with 20 items, each item is [Dataset_Dependent]"*. Here, *Dataset_Dependent* acts as a flexible placeholder, tailored according to the dataset. For instance, we use *"a movie review"* for SST-2 and *"a sentence gathered from news articles. These sentences contain topics including World, Sports, Business, and Technology."* for AGNews. By invoking this prompt a hundred times, we produce a total of 2,000 GPT-crafted inputs for each dataset.

After obtaining the outputs from the target model using the probing dataset, we harness these results to train surrogate/replica models using LoRA. To assess the success rate of our model-stealing approach, we adopt a matching score called "agreement" (Jagielski et al., 2020). This metric allows for a direct comparison between the outputs of the target and surrogate models for each sample, providing a reliable measure of the functional similarity between the two models. A match, irrespective of the correctness of the output, is considered a success. In addition, we calculate the accuracy of the surrogate models. Given the observed consistency between accuracy and agreement, we relegate the accuracy results to Appendix D and base our analysis of performance primarily on the agreement metric.

### 4.3 EVALUATION SETTINGS

We follow the same evaluation settings as the one of membership inference (Section 3.3), specifically, models fine-tuned by the different adaptation techniques that achieve comparable performance.

The surrogate model undergoes fine-tuning from an identical base model, utilizing LoRA with the specified parameters: `r=16`, `lora_alpha=16`, `lora_dropout=0.1`, `bias=all`. This fine-tuning is performed over five epochs, with a learning rate determined at $1 \times 10^{-3}$. For every target model under consideration, the experiments are replicated five times, each instance employing a distinct random seed.

### 4.4 RESULTS

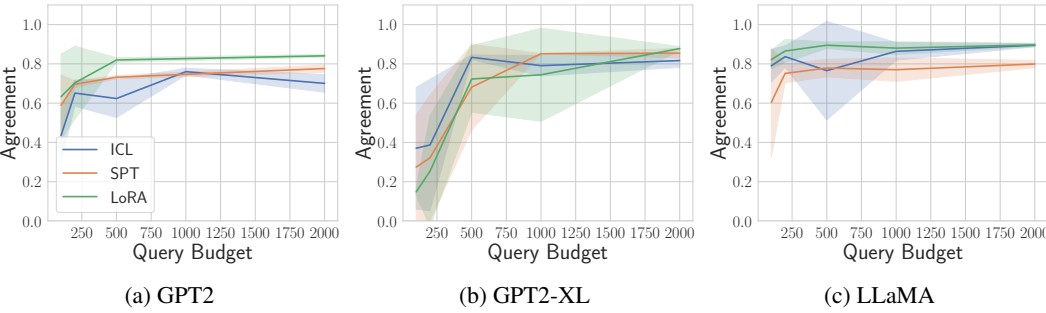

(a) GPT2         (b) GPT2-XL         (c) LLaMA

Figure 5: Model stealing performance across various query budgets for DBPedia-trained models.

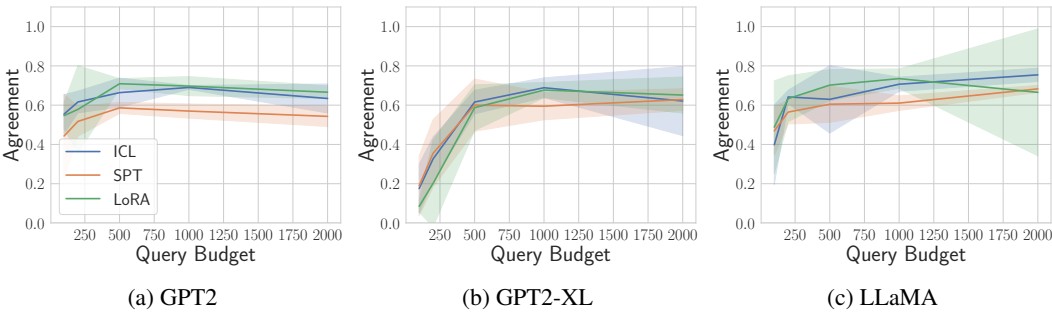

Figure 6: Model stealing performance for DBPedia-trained models using GPT3.5-generated data.

We initiate our assessment of the model stealing attack by examining various query budgets, i.e., probing datasets with different sizes. For this evaluation, we employ the DBPedia dataset and draw samples for the probing datasets from the same distribution as the dataset of the target model. The results, illustrated in Figure 5, indicate that even with a constrained set of queries, the surrogate model aligns closely with the target model. For example, for all three model sizes, a mere 1,000 samples suffice to replicate a surrogate model that mirrors over 80% of the target's functionality. It is crucial to highlight that these unlabeled samples (that are subsequently labeled using the target model) are substantially more cost-effective to obtain compared to the labeled data used in the fine-tuning of the target model.

We next assess the same settings but with a more lenient assumption, wherein the adversary lacks data from the target distribution. Instead, GPT-generated data is employed for constructing the probing dataset. As depicted in Figure 6, using such artificially generated data yields results comparable to those from the same distribution. This contrasts with vision tasks where replicating an image classification model requires a substantially larger query budget without access to data from the same distribution (Liu et al., 2022; Truong et al., 2021).

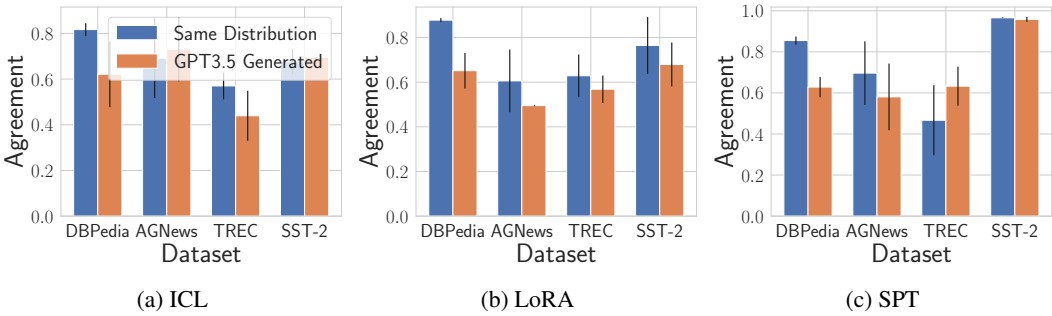

Figure 7: Comparative analysis of model stealing attacks on GPT2-XL-based models: examining the impact of different probing dataset sources.

To further compare the performance of using generated data and data from the same distribution, we fix the query budget at 2,000 and assess the performance across the four datasets with GPT2-XL, as depicted in Figure 7. As expected, using data from the same distribution is better, however, for most of the cases, the difference is marginal. This trend is consistent across various model architectures, as demonstrated in the results presented in Appendix D. Intriguingly, there are instances, such as with AGNews (Figure 7a) and TREC (Figure 7c), where generated data actually facilitates a more successful model stealing attack. This observation opens the door to the potential of enhancing such attacks by optimizing data generation—perhaps leveraging sophisticated prompts or superior generation models—a direction we aim to explore in subsequent work.

In conclusion, our findings emphasize the vulnerability of all three fine-tuning methods to model stealing attacks, even when the adversary has a limited query budget and lacks access to the target model's training data distribution.

## 5 BACKDOOR ATTACK

Lastly, we investigate an additional security threat against ICL, LoRA, and SPT: the backdoor attack. This attack occurs during training when an adversary poisons the training dataset of a target model to introduce a backdoor. This backdoor is associated with a trigger such that when an input possesses this trigger, a particular output, as designated by the adversary, is predicted. This output might be untargeted, where the aim is merely an incorrect prediction, or it can be targeted to yield a specific label chosen by the adversary. In this work, we focus on the later –more complex– case, i.e., the targeted backdoor attack.

### 5.1 THREAT MODEL

We follow previous backdoor attacks (Gu et al., 2017) threat model and make no specific assumptions about the target model other than its vulnerability to having its fine-tuning dataset poisoned. It is important to recap that the term "fine-tuning dataset" in this context pertains to the data leveraged by ICL, LoRA, and SPT for adapting the target model.

### 5.2 METHODOLOGY

To execute the backdoor attack, we start by crafting a backdoored dataset. First, we sample a subset from the fine-tuning dataset and integrate the trigger into every input. Next, we switch the associated label to the predetermined –backdoor– target label. For the purposes of this study, this label is set to 0. Once the backdoored dataset is ready, it is merged with the clean fine-tuning dataset, and then the target models are trained using the respective techniques. We do not replace clean samples but concatenate the fine-tuning dataset with the backdoored one.

For evaluation, we follow previous backdoor attack works (Gu et al., 2017; Salem et al., 2022; Kandpal et al., 2023) that use two primary metrics: utility and attack success rate. Utility quantifies the performance of the backdoored model using a clean test dataset. The closer this metric aligns with the accuracy of an unaltered –clean– model, the more effective the backdoor attack. The attack success rate, on the other hand, evaluates how accurately backdoored models respond to backdoored data. We construct a backdoored test dataset by inserting triggers into the entirety of the clean test dataset and reassigning the label to our target value (i.e., 0), and then use this dataset to evaluate the backdoored model. An attack success rate of 100% represents a perfect backdoor attack's performance.

Finally, in the ICL scenario, given that the count of examples is constrained, we ensure that the backdoored dataset excludes any inputs whose original label coincides with the target label. This aims to maximize the performance of the backdoor attack in the ICL settings. Furthermore, acknowledging the influence of demonstration order on ICL performance (Zhao et al., 2021), we adopt two separate poisoning approaches for ICL. In the first approach, we poison sentences at the start of the prompt, and in the second, we target sentences at the prompt's end.

### 5.3 EVALUATION SETTINGS

We follow the same evaluation settings as the one of membership inference (Section 3.3), but with the added step involving the creation of a backdoored fine-tuning dataset before initiating model training. We construct the backdoored fine-tuning dataset as follows: For each selected clean sentence, we introduce the trigger word *"Hikigane"* (which translates to "trigger" in Japanese) at its beginning and adjust its associated label to class 0. These modified sentences are then added to the clean fine-tuning dataset without removing any original samples.

We assess the backdoor attack across varying poisoning rates. Specifically, for LoRA and SPT, the poisoning rate ranges between 0.1 and 0.75. For ICL, given that we use only four demonstrations, we examine scenarios with 1, 2, or 3 poisoned demonstrations, resulting in poisoning rates of 0.25, 0.5, and 0.75, respectively.

### 5.4 RESULTS

We first assess the backdoor attack across varying poisoning rates using the three datasets: DBPedia, AGNews, and TREC with the GPT2-XL model. The results are illustrated in Figure 8. From our

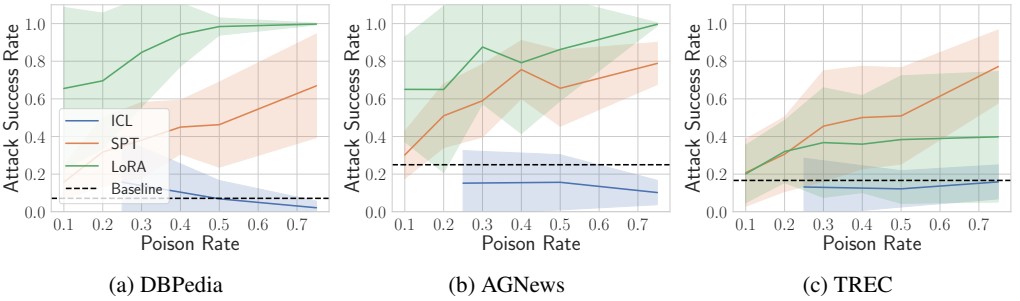

Figure 8: Comparison of attack success rates at different poison rates for GPT2-XL models.

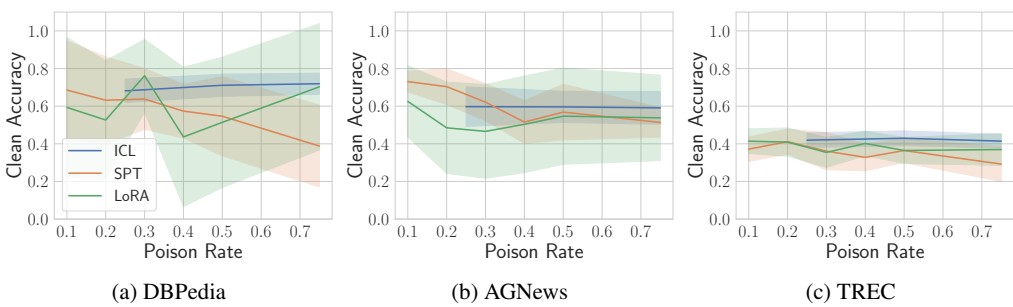

Figure 9: Comparison of utility at different poison rates for GPT2-XL models.

preliminary experiments, we decided to omit the SST-2 dataset. Since its binary structure, when subjected to a backdoor, substantially reduced the model utility across all adaptation methods.

As anticipated, for LoRA and SPT, an increase in the poisoning rate boosts the attack success rate (ASR) of the backdoor attack. This rise can be attributed to the model's improved trigger recall as it encounters more backdoored data during the fine-tuning. Conversely, the utility of the backdoored model sees a minor decline as the poisoning rate grows, as shown in Figure 9. This could be a result of the model slightly overfitting to the backdoored pattern, possibly weakening the connection between clean sentences and their designated classes

Conversely, In-Context Learning (ICL) shows minimal variation in performance as the poison rate increases, consistently approximating random guessing. We speculate that the limited number of demonstrations might cause this, making the model rely more on its inherent knowledge rather than the backdoored new input. Kandpal et al. (2023) explores a situation where backdooring takes place before model adaptation through ICL, i.e., the model is first fine-tuned with backdoored data. Their findings indicate robust backdoor performance, even in the absence of backdoored demonstrations. This aligns with our hypothesis that ICL models draw more from their inherent knowledge than from the few provided demonstrations.

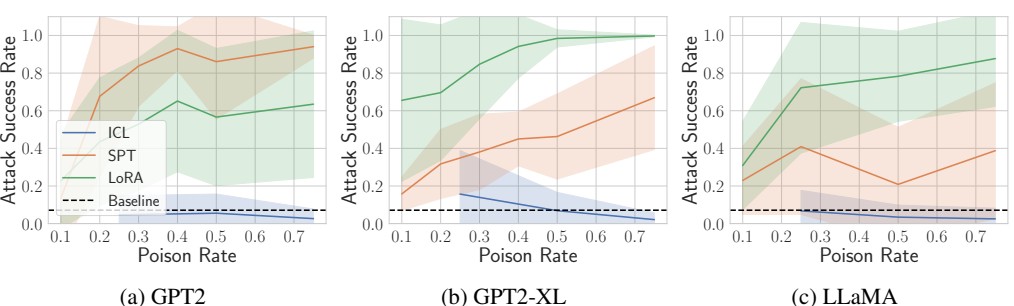

Figure 10: Comparison of attack success rates at various poison rates for DBPedia models.

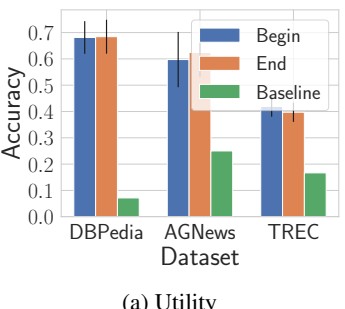 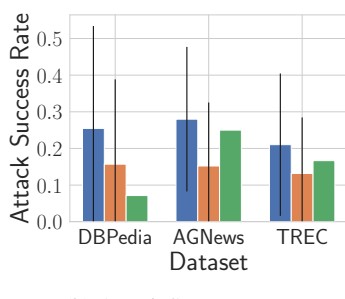

(a) Utility                      (b) Attack Success Rate

Figure 11: Backdoor attack performance when poisoning the first or the last demonstration in the prompt. The baseline indicates random guessing performance for the –target– label 0.

Our observation extends to models of varying sizes. As shown in Figure 10, ICL exhibits an ASR close to random guessing across all three models, while SPT and LoRA consistently outperform ICL by a significant margin. We further validate the transferability of our conclusion for different target labels, as shown in Appendix C.

Finally, we investigate whether poisoning either the first or the demonstration in the prompt yields a noticeable difference. To this end, we independently poison the first and last demonstration in the prompt and plot the results in Figure 11. The results indicate a marginal increase in attack success rate when the initial sentence is poisoned, even though the variation is minimal. These results show that the location of poisoned data within the prompt does not substantially influence the effectiveness of the backdooring approach in the context of ICL.

## 6 DISCUSSION AND LIMITATIONS

While we recognize that more advanced attacks could target Language Models (LLMs), especially in pretrained or full fine-tuning scenarios, our study serves as an empirical lower bound for evaluating vulnerabilities across diverse LLM adaptation techniques. Our findings highlight the inherent vulnerabilities of these techniques to a variety of threats, emphasizing the pressing need for robust defenses in such settings.

To the best of our knowledge, the majority of defenses against privacy and security threats are tailored for full fine-tuning scenarios. However, we believe that the core of these defenses can be adapted to the LLM adaptation techniques. For instance, recent works have successfully extended differential privacy, a well-established defense with guarantees against membership inference attacks, to ICL settings (Panda et al., 2023; Duan et al., 2023b; Tang et al., 2023). Moving forward, we intend to adapt these defenses to the LLM adaptation techniques and assess their efficacy against the presented attacks.

## 7 CONCLUSION

In this study, we have systematically investigated the vulnerabilities of existing adaptation methods for Large Language Models (LLMs) through a three-fold assessment that encompasses both privacy and security considerations. Our findings reveal three key insights into the security and privacy aspects of LLM adaptation techniques. Firstly, In-Context Learning (ICL) emerges as the most vulnerable to membership inference attacks (MIAs), underscoring the need for enhanced privacy defenses in the implementation of this technique. Secondly, our study reveals a pervasive vulnerability across all three training paradigms to model stealing attacks. Intriguingly, the use of GPT3.5-generated data demonstrates a strong performance in such attacks, highlighting the ease with which fine-tuned LLMs can be stolen or replicated. Lastly, with respect to backdoor attacks, our results indicate that Low-Rank Adaptation (LoRA) and Soft Prompt Tuning (SPT) exhibit a higher susceptibility, whereas ICL proves to be less affected. These insights emphasize the necessity for tailored defenses in the deployment of LLM adaptation techniques. Moreover, they underscore each technique's vulnerabilities, alerting users to the potential risks and consequences associated with their use.

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

Table 1: Examples of the prompts used for text classification for the ICL setting.

| Task | Prompt | Label Names |
|---|---|---|
| DBPedia | Classify the documents based on whether they are about a Company, School, Artist, Athlete, Politician, Transportation, Building, Nature, Village, Animal, Plant, Album, Film, or Book. | Company, School, Artist, Athlete, Politician, Transportation, Building, Nature, Village, Animal, Plant, Album, Film, Book |
| | Article: Leopold Bros. is a family-owned and operated distillery located in Denver Colorado. Answer: Company | |
| | Article: Aerostar S.A. is an aeronautical manufacturing company based in Bacău Romania. Answer: | |
| AGNews | Article: Kerry-Kerrey Confusion Trips Up Campaign (AP),"AP - John Kerry, Bob Kerrey. It's easy to get confused." Answer: World | World, Sports, Business, Technology |
| | Article: IBM Chips May Someday Heal Themselves,New technology applies electrical fuses to help identify and repair faults. Answer: | |
| TREC | Classify the questions based on whether their answer type is a Number, Location, Person, Description, Entity, or Abbreviation. | Number, Location, Person, Description, Entity, Abbreviation |
| | Question: What is a biosphere? Answer Type: Description | |
| | Question: When was Ozzy Osbourne born? Answer Type: | |
| SST-2 | input: sentence - This movie is amazing! output: Positive; | Positive, Negative |
| | input: sentence - Horrific movie, don't see it. output: | |

## A  MODEL PERFORMANCE AND TRAINING HYPERPARAMETERS

### A.1  MODEL PERFORMANCE

As outlined in Section 3.1, careful management of the training dataset size and training hyperparameters has been undertaken to ensure that both SPT and LoRA exhibit accuracy levels comparable to ICL. Consequently, this section exclusively presents the performance metrics for ICL across various tasks.

For SST-2, the model attains an accuracy of approximately 85%. In the case of DBPedia, AGNews, and TREC, the model demonstrates accuracies of about 70%, 70%, and 45%, respectively. Notably, these findings align with those reported in prior research by Zhao et al. (2021).

### A.2  HYPERPARAMETERS

**ICL:** ICL involves appending the input to a predetermined prompt, constructed with four demonstrations and accompanying illustrative words. The prompt formatting adheres to the conventions outlined by Zhao et al. (2021), with some examples provided in Table 1.

**LoRA:** We set the LoRA configuration to `r=16`, `lora_alpha=16`, `lora_dropout=0.1`, `bias="all"`. The model is fine-tuned over five epochs, employing a learning rate of $1 \times 10^{-3}$. To

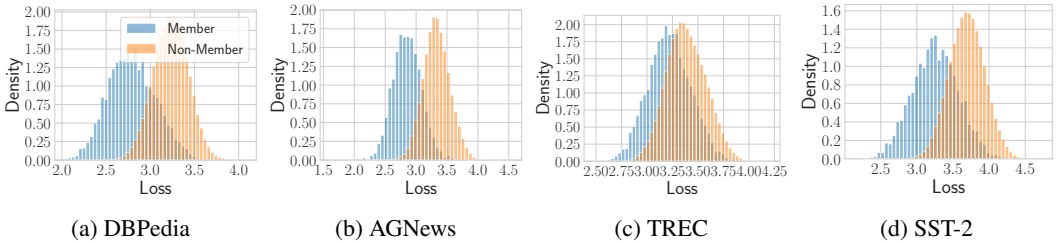

(a) DBPedia     (b) AGNews     (c) TREC     (d) SST-2

Figure 12: Loss distribution for member and nonmember samples using GPT2-XL.

ensure a comparable performance with ICL, the fine-tuning process utilizes 300, 200, 300, and 600 samples for the DBPedia, AGNews, TREC, and SST-2 datasets, respectively.

**SPT:** For SPT, the number of virtual tokens is set to ten. The model undergoes fine-tuning for five epochs, with a learning rate of $3 \times 10^{-3}$. Similar to LoRA, the fine-tuning samples are adjusted to ensure a performance benchmark consistent with ICL. Specifically, 800, 200, 900, and 1000 samples are used for the DBPedia, AGNews, TREC, and SST-2 datasets, respectively.

## B   LOSS DISTRIBUTION

We depict the loss distribution for both member and nonmember samples in Figure 12. The figure illustrates a statistically significant trend, with member samples consistently exhibiting lower loss values compared to nonmember samples.

## C   BACKDOOR ATTACK AGAINST DIFFERENT TARGET CLASS

We conduct the backdoor attack with a different target class (class one), and experimental results confirm the stability of the previously reached conclusion. Specifically, across different model architectures, as illustrated in Figure 13, SPT and LoRA consistently exhibit superior performance in conducting attacks compared to ICL.

## D   MODEL STEALING

We focus on the DBPedia-trained models, and present a figure illustrating the variation in accuracy corresponding to different query budgets in Figure 14. Notably, we observe a nearly identical trend in accuracy compared to the agreement results. Additionally, we extend our analysis to include the use of GPT3.5-generated data for model stealing, and the performance of the surrogate model is illustrated in Figure 15.

Furthermore, we explore the impact of using data from different sources, as delineated in Figure 16. Our findings consistently indicate that, irrespective of model architectures, querying with data from

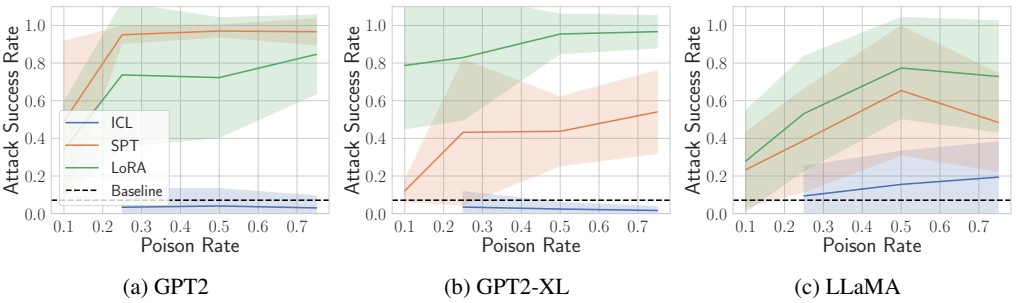

(a) GPT2     (b) GPT2-XL     (c) LLaMA

Figure 13: Backdoor performance with the target label 1 on the DBPedia dataset.

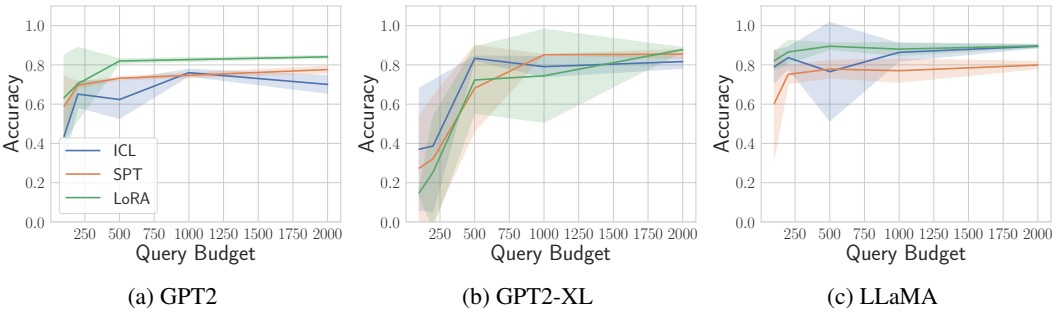

(a) GPT2        (b) GPT2-XL        (c) LLaMA

Figure 14: Performance (accuracy) of model stealing with probing data from the same distribution, across different query budgets for models trained on DBPedia.

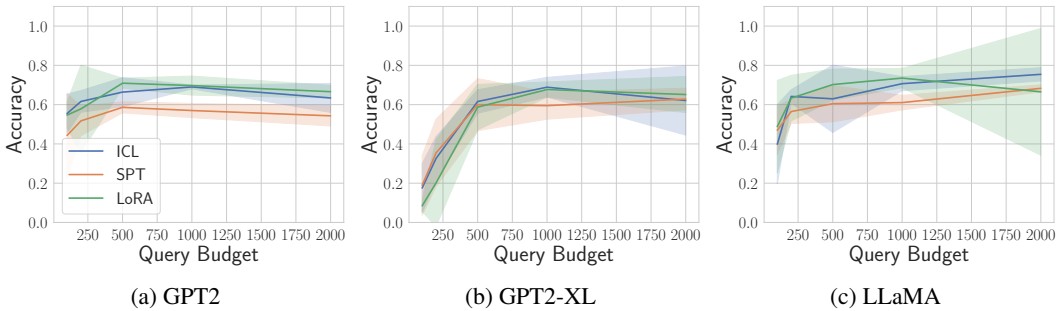

(a) GPT2        (b) GPT2-XL        (c) LLaMA

Figure 15: Performance (accuracy) of model stealing with GPT3.5-generated as the probing data, across different query budgets for models trained on DBPedia.

the same distribution consistently outperforms querying with GPT3.5-generated data, albeit with a modest difference in performance for many cases.

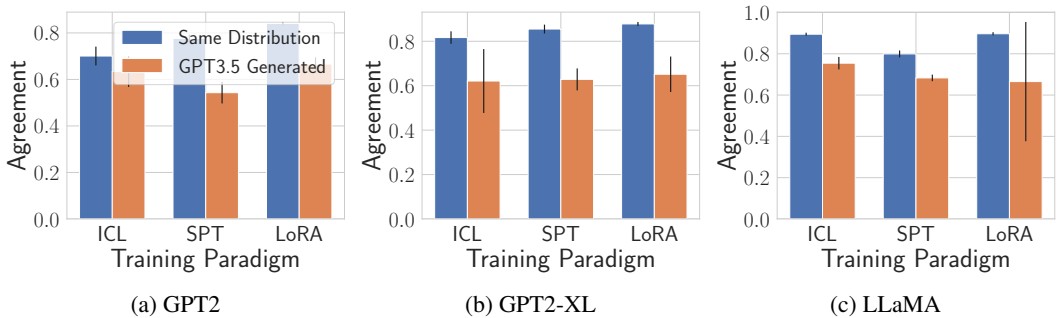

(a) GPT2          (b) GPT2-XL          (c) LLaMA

Figure 16: Comparison of the model stealing attack on various model architectures using the DBPedia dataset.

