# OpenReview forum: "Last One Standing: A Comparative Analysis of Security and Privacy of Soft Prompt Tuning, LoRA, and In-Context Learning"
_ICLR.cc/2024/Conference — Submitted to ICLR 2024_

### Official Review · Reviewer_J1hc · 2023-10-31

**Soundness:** 3 good
**Presentation:** 4 excellent
**Contribution:** 2 fair
**Rating:** 5
**Confidence:** 3

**Summary:**

In this work, a comparison of three LLM adaptation methods is provided with regards to their resiliency against three privacy and security threats. As one might expect, no single technique emerges as a clear winner in terms of robustness but different techniques seem to have certain strengths and weaknesses.

**Strengths:**

1. The presentation, writing, and results in the paper (for example, Figure 1 to visualize the trade-off) are very clear and easy to follow.
2. The comparison seems comprehensive in the sense that it covers multiple different LLM architectures, different adaptation techniques, and multiple datasets.

**Weaknesses:**

I think that the while the paper claims to make assessments on security and privacy, the three attacks considered here are not comprehensive enough. For example, just on the basis of resiliency results again one backdoor attack and one model stealing attack, it might be a bit preemptive to make generalized statements regarding security. In particular, prompt injection attacks are a major security threat which do not seem to be covered here at all. Similarly, for privacy, while membership inference attack is one kind of threat, other relatively sophisticated privacy attacks pertaining to PII memorization/extraction can be crafted.

**Questions:**

1. While conducting the MIA, the classification is done based on loss function value. So is the assumption that the adversary knows the correct loss function?
2. Also, what significance does the loss function value have for ICL exemplars? For LoRA and SPT, the LLM has learned to minimize loss on member exemplars, but I do not know what relationship the ICL samples have with the loss function value.
3. In section 3.2, it is not clear what is meant by the sentence "Subsequently, we calibrate the hyperparameters of LoRA and SPT to align their performance with that of ICL." What exactly are the steps taken to ensure that the comparison of these methods is apples-to-apples? How can adding, say, 4 ICL examples be equated to performing LoRA with hundreds of training samples?
4. In the model stealing, the success of the attack is quantified by a certain "agreement" score between the target model and a surrogate model. Can the authors please define what this metric exactly is, as it is not available anywhere.

---

> ### Author Response · Authors · 2023-11-14
> **Response to Reviewer J1hc**
>
> > Regarding Prompt injection attacks
>
> We agree with the reviewer that prompt injection attacks are a current threat. However, our focus is on the fine-tuning scenario. The prompt injection attacks, however, are more closely associated with the vulnerabilities of Language Model Models (LLMs) rather than the fine-tuning techniques themselves. Existing models have been demonstrated to be susceptible to such attacks, and as of now, there are no established successful defenses against them. The fine-tuning techniques we currently employ are tailored for benign tasks and are not directly related to mitigating prompt injection attacks. Nevertheless, it would indeed be intriguing to investigate whether these fine-tuning methods could potentially increase or decrease the models' vulnerability to such attacks. We aim to explore this in future research.
>
> > Regarding PII leakage
>
> We acknowledge the significance of these PII leakage attacks and agree that they are indeed interesting. However, we would like to point out that the Membership Inference Attack (MIA) represents the most basic form of privacy attack. As such, it provides an overview of more sophisticated attacks and is even shown to imply different privacy attacks[1]. Similarly, current works analyzing the differential privacy guarantee use MIA only and not the other different privacy attacks[2,3].
>  For example, if a technique, like In-Context Learning (ICL), is more vulnerable to MIA, then it is likely to be more vulnerable to other forms of vulnerabilities, such as Personally Identifiable Information (PII) leakage, compared to methods like LoRA and SPT.
>
> [1] SoK: Let the Privacy Games Begin! A Unified Treatment of Data Inference Privacy in Machine Learning
>
> [2] Bayesian Estimation of Differential Privacy
>
> [3] Adversary instantiation: Lower bounds for differentially private machine learning
>
> > While conducting the MIA, the classification is done based on loss function value. So is the assumption that the adversary knows the correct loss function?
>
> We do not make any assumption on the correct loss function. Our approach simply requires that the model generates a list of candidate words along with their respective probabilities for a given test sample. This allows the attacker to compute the loss using their chosen loss function, based on the provided information, without the need to know the loss function utilized internally by the model.
>
> > Also, what significance does the loss function value have for ICL exemplars? For LoRA and SPT, the LLM has learned to minimize loss on member exemplars, but I do not know what relationship the ICL samples have with the loss function value.
>
> It seems there may be a bit of a misunderstanding regarding our approach. In the context of the Membership Inference Attack (MIA), the calculation of the loss occurs exclusively during the testing phase. The focus is on assessing the loss during the model's testing, which allows us to infer membership status (without depending on the loss function used in training).
>
> Should there be any need for further clarification, or if our understanding of your question has missed the mark, please let us know.
>
> > In section 3.2, it is not clear what is meant by the sentence "Subsequently, we calibrate the hyperparameters of LoRA and SPT to align their performance with that of ICL." What exactly are the steps taken to ensure that the comparison of these methods is apples-to-apples? How can adding, say, 4 ICL examples be equated to performing LoRA with hundreds of training samples?
>
> Our calibration process is designed to evaluate the privacy and security risks of different methods when they exhibit comparable performance.
>
> To establish this equivalence, we concentrate on accuracy. For LoRA and SPT, we carefully modify the number of training samples to match their accuracy levels with that of In-Context Learning (ICL) for the same task. This approach ensures that we compare the privacy and security impacts associated with these different adaptation methods at similar or identical levels of accuracy. We will clarify this in our paper.
>
> > In the model stealing, the success of the attack is quantified by a certain "agreement" score between the target model and a surrogate model. Can the authors please define what this metric exactly is, as it is not available anywhere.
>
> We introduce the agreement score in Section 4.2. Intuitively, this metric compares the outputs generated by the target and surrogate models for each sample. A 'success' is registered when the outputs from both models are similar, regardless of whether these outputs are correct or not. The agreement score is subsequently calculated as the ratio of successful matches to the total number of samples in the testing dataset.

---

> ### Author Response · Authors · 2023-11-22
>
> We thank the reviewer again for their feedback and would like to ask if there is anything else that requires clarification. As previously mentioned, membership inference attack is the most common and fundamental privacy attack, which other more advanced attacks, such as reconstruction attacks, can be built upon. Therefore, if a model is secure against membership inference attacks, it is also secure against reconstruction attacks. Regarding other security attacks, such as prompt injection attacks or jailbreaks, there are so far no peer-reviewed papers available, and the arXiv ones require large LLMs that may not be applicable to some of the models we studied in this work. Nonetheless, we agree that this is an interesting direction that we would like to explore further in future research.

---

> > ### Comment · Reviewer_J1hc · 2023-11-22
> >
> > Thanks for your response to my questions. After going through your answers, I have updated my score accordingly.

---

### Official Review · Reviewer_RTpz · 2023-10-31

**Soundness:** 3 good
**Presentation:** 3 good
**Contribution:** 3 good
**Rating:** 8
**Confidence:** 4

**Summary:**

The paper investigates the susceptibility of fine-tuning techniques on LLMs to membership inference attacks, model stealing attacks and backdoor attacks. In the experimental evaluation, Low-Rank Adaption (LoRA), Soft Prompt Tuning (SPT) and In-Context Learning (ICL) are evaluated against these three types of attacks. The results show that not every fine-tuning method is equally susceptible to all three attacks. For example, while ICL is mostly resilient against backdoor attacks, membership inference attacks are very successful against models fine-tuned using ICL. Overall, it seems that each of these fine-tuning methods have their pros and cons regarding the susceptibility to security and privacy attacks. The paper raises awareness of these risks when using these fine-tuning methods.

**Strengths:**

- The paper is well structured, well written and easy to follow
- The fact that different fine-tuning methods are not equally susceptible to privacy and security attacks is intriguing
- The study of the privacy and security aspects of different fine-tuning methods seems to be novel and timely as LLMs are often not trained from scratch, but instead fine-tuned

**Weaknesses:**

- The backdoor attack is only evaluated with the target label 0. The results might differ for targeting other classes, as other classes might be more difficult to predict using a backdoor.
- For the backdoor attacks, it would be nice to draw the baseline of random guessing into the attack success rate plots (Fig. 8 and Fig. 10). This allows the reader to quickly see how effective the backdoor attack is.
- With the limited context length of the LLMs it is hard to make a definitive conclusion about the susceptibility of ICL to membership inference attacks. There seems to be a trend that increasing the number of given examples reduces the susceptibility. However, the authors acknowledge this trend.

Misc:
- Page 9: missing word - "Finally, we investigate whether poisoning either the first or the [last] demonstration [...]"

**Questions:**

- Q1: How was the membership inference attack on ICL performed? And was the whole prompt/all of the given examples considered as the "member" here?
- Q2: What are the accuracies of the original fine-tuned models using ICL, SPT and LoRA on the different datasets?
- Q3: Could you provide experimental results for backdoor attacks with different target labels other than the label 0?
- Q4: "We do not replace clean samples but concatenate the fine-tunig dataset with the backdoored one" -> Does this mean that samples are present twice in the fine-tuning dataset? Once with the backdoor trigger and once without the backdoor trigger?
- Q5: Why is the standard deviation of the backdoor attacks so high?

---

> ### Author Response · Authors · 2023-11-14
> **Response to Reviewer RTpz**
>
> > For the backdoor attacks, it would be nice to draw the baseline of random guessing into the attack success rate plots (Fig. 8 and Fig. 10). This allows the reader to quickly see how effective the backdoor attack is.
>
> Thanks for the suggestion. We have updated the revised version accordingly.
>
> > Q1: How was the membership inference attack on ICL performed? And was the whole prompt/all of the given examples considered as the "member" here?
>
> We consider each example within the prompt to be treated as a distinct member. During the membership inference attack, we input each tested sample into the language model and leverage the loss to ascertain its membership status. We will clarify this in the paper.
>
> > Q2: What are the accuracies of the original fine-tuned models using ICL, SPT, and LoRA on the different datasets?
>
> In our settings, we adjust the size of the training dataset such that different adaptation methods demonstrate similar performance. We have included the accuracy in our updated paper.
>
> > Q3: Could you provide experimental results for backdoor attacks with different target labels other than the label 0?
>
> This is an interesting idea, we will run this experiment in the following days.
>
> > Q4: "We do not replace clean samples but concatenate the fine-tuning dataset with the backdoored one" -> Does this mean that samples are present twice in the fine-tuning dataset? Once with the backdoor trigger and once without the backdoor trigger?
>
> Indeed, your interpretation is correct. Our decision is tailored to the constraints of In-Context Learning (ICL), which can handle only a limited number of demonstrations. By strategically including a mix of both clean and poisoned samples, our goal is to minimize potential confusion for the language model during its training process, thereby enhancing its performance.
>
> > Q5: Why is the standard deviation of the backdoor attacks so high?
>
> To align the performance of LoRA and SPT with that of ICL, we limit the number of training samples. Consequently, the selection and poisoning of the training dataset introduce a notable degree of randomness, significantly impacting final performance outcomes.
>
> To mitigate this variability and guarantee the reliability of our results, we have conducted each experiment ten times. This approach helps to provide a more representative average performance across the experiments.

---

> > ### Comment · Reviewer_RTpz · 2023-11-17
> > **Reply to Author Rebuttal**
> >
> > Thank you very much for the detailed answers to my questions.
> >
> > Is it maybe possible to post the results of the experiment in Q3 or the accuracies of the models here as a response? Or maybe update the version of the paper with a more recent revision?
> >
> > Having this additional information, I would be willing to raise my score to "accept".

---

> > > ### Author Response · Authors · 2023-11-21
> > > **Thank you for your response**
> > >
> > > We thank the reviewer for his support and suggestions. We ran experiments to test the backdoor performance on a different target label, i.e., 1, and have included the results in Appendix C.
> > >
> > > In general, the results confirm our earlier conclusion that SPT and LoRA consistently perform better than ICL in attacking. But, interestingly, when we looked at large language models like LLaMA, we noticed that the backdoor's success rate varied based on the chosen target label.
> > >
> > > For instance, for the ICL case, targeting label 1 yielded a significantly higher attack success rate (19.4%) compared to label 0 (2.5%). When we tested on label 2, the success rate dropped to 0%.  This indicates that the backdoor performance for LLM may indeed link with the trigger design, which encourages further research to figure out a better way to compromise ICL, like maybe trying a prompt tuning approach but focusing on optimizing the demonstrations. We thank the reviewer again for guiding us to this observation and are happy to provide more explanations or experiments if needed.

---

> > > > ### Comment · Reviewer_RTpz · 2023-11-22
> > > > **Thanks for the clarification**
> > > >
> > > > Thank you for updating the paper and answering my questions.
> > > >
> > > > My questions have been appropriately addressed, which is why I am raising my score to "accept".

---

> ### Author Response · Authors · 2023-11-22
> **Thank you for the suggestions**
>
> Thanks again for your suggestions to improve our work.

---

### Official Review · Reviewer_FB1d · 2023-10-31

**Soundness:** 1 poor
**Presentation:** 3 good
**Contribution:** 1 poor
**Rating:** 1
**Confidence:** 4

**Summary:**

This paper evaluates the security and privacy risks of three adaptation methods, namely LoRA, soft prompt tuning, and in-context learning. The paper evaluates model privacy based on a loss-based membership inference attack and a model stealing attack. The model security is evaluated based on a backdoor attack.

**Strengths:**

1.	Security and privacy investigated in the paper are both critical for LoRA, soft prompt tuning, and in-context learning.
2.	This paper conducts the first comprehensive analysis on adaptation techniques, filling a gap in the existing literature in terms of security and privacy analysis.
3.	Figure 1 provides a good summary of the overall evaluation results.

**Weaknesses:**

1.	My major concern is the outdated attacks used in the evaluation. Most evaluations are conducted based on out-of-date and less effective attacks. For example, the paper uses only a loss-based membership inference attack to evaluate the training data privacy, which has been shown ineffective in the existing work. Recent and more advanced attacks like LiRA [1] should be used. Similarly recent model stealing and backdoor attacks are not evaluated in the paper.
2.	The paper only reports the evaluation results. However, the in-depth analysis of the results is missing. For example, why do LoRA and SPT achieve good privacy performance against membership inference attacks? Does the number of training samples or the number of training iterations play an important role in security/privacy risk? More in-depth ablations would benefit the paper.
3.	The security and privacy evaluation results do not include the accuracy results, which is insufficient. Typically, there exists a trade-off between accuracy and security/privacy.
4.	It would be interesting to compare three adaptation techniques with a naïve fine-tuning approach regarding the security and privacy risks.

[1] Carlini, Nicholas, Steve Chien, Milad Nasr, Shuang Song, Andreas Terzis, and Florian Tramer. "Membership inference attacks from first principles." In 2022 IEEE Symposium on Security and Privacy (SP), pp. 1897-1914. IEEE, 2022.

**Questions:**

Please provide a more in-depth analysis of the evaluation and the accuracy achieved by all the evaluated models.

---

> ### Author Response · Authors · 2023-11-14
> **Response to Reviewer FB1d**
>
> > Recent and more advanced attacks like LiRA should be used.
>
> We appreciate your suggestion to incorporate LiRA into our study, given its strong performance in MIA. However, its application seems impractical for the language Model Models (LLMs), particularly due to the computational constraints of training **tens of thousands** of shadow models. In fact, extending LiRA to large-scale LLMs is substantial enough to be its own paper. We encourage a closer examination of the model sizes used in our study compared to those in LiRA's research. Conducting such experiments on large LLMs requires a budget that is beyond the reach of many, a point underscored by **even industry giants like Google, who prefer simpler models like ResNet-9 for their quick training capabilities and computational efficiency [1]**.
>
> Moreover, while the attack method used in our paper may be simple, it significantly reveals membership status vulnerabilities in the ICL. This finding should prompt users considering ICL to weigh their options more carefully. Employing more advanced attacks, in this context, would likely yield no further insights.
>
> [1] Students Parrot Their Teachers: Membership Inference on Model Distillation
>
> > Similarly recent model stealing and backdoor attacks are not evaluated in the paper.
>
> We wish to emphasize that the focus of recent model stealing attacks[2][3][4], is primarily on executing these attacks under more constrained conditions, such as limited knowledge of dataset distribution or access only to hard-label predictions. In contrast, our approach provides the attacker with the most extensive information possible, offering an upper-bound assessment of the model's susceptibility to stealing.
>
> For backdoor attacks, due to the nature of in-context learning, there are limited operations we can do except poisoning the demonstrations. Therefore, we are curious about the specific types of attacks the reviewer envisions, so we can evaluate their feasibility in our context.
>
> Lastly, it's important to acknowledge our commitment to reducing our carbon footprint and promoting green computing. In line with this commitment, we aim to minimize unnecessary experiments. By simulating scenarios where the adversary is given optimal conditions, we focus our efforts on safeguarding against the most severe potential threats.
>
> [2] Data-Free Model Extraction
>
> [3] Towards Data-Free Model Stealing in a Hard Label Setting
>
> [4] MAZE: Data-Free Model Stealing Attack Using Zeroth-Order Gradient Estimation
>
> > In-depth analysis of the results is missing…Does the number of training samples or the number of training iterations play an important role in security/privacy risk?
>
> While we appreciate the reviewer's inquiry, it appears that the question raised might have been overlooked in our paper. We would respectfully suggest revisiting Section 3.4 of our paper, where we have extensively discussed this very topic. To understand the impact of training sample quantity on the Membership Inference Attack (MIA), we conducted thorough assessments with varied sample counts, including 4 and 8 demonstrations. These specific findings, which directly address the reviewer’s concern, are presented in Figure 4.
>
> > do not include the accuracy results, which is insufficient. Typically, there exists a trade-off between accuracy and security/privacy.
>
> We acknowledge the inherent trade-off between accuracy and security/privacy, therefore in our study, as outlined in Section 3.3, our methodology involves adjusting the training dataset size to ensure comparable performance across various adaptations. This approach is in line with established practices in previous research [5]. We also agree that including the accuracy would be necessary, we have updated our paper accordingly.
>
> [5] On the Privacy Risk of In-context Learning
>
> > Compare three adaptation techniques with a naïve fine-tuning approach
>
> Indeed, that would be interesting. However, the primary reason we chose to compare these three adaptation methods stems from the computational inefficiency of fine-tuning models. In practice, there's a noticeable trend toward utilizing more efficient approaches. Our study aims to investigate the privacy and security vulnerabilities associated with these efficient adaptation methods, shedding light on their potential implications in real-world scenarios.

---

> ### Author Response · Authors · 2023-11-22
>
> We thank the reviewer again for their feedback and would like to ask if there is anything else that requires clarification. Furthermore, we would like to emphasize that conducting LiRA on LLMs is not feasible computationally, as it would require at least 100 A100s over the course of three weeks. We were unable to find any prior instances of LiRA on LLMs, even with Google using smaller models like ResNet-9.

---

### Official Review · Reviewer_JTKU · 2023-11-01

**Soundness:** 3 good
**Presentation:** 3 good
**Contribution:** 2 fair
**Rating:** 5
**Confidence:** 3

**Summary:**

The paper studies the privacy and security property of several parameter-efficient fine-tuning methods using 3 attacks.

**Strengths:**

1. The paper is the first to systematically study the privacy and security effect of parameter-efficient fine-tuning methods.
2. The paper is well-written and easy to follow.

**Weaknesses:**

1. I am mainly concerned that the results are not generalizable since these properties can heavily depends on the model architecture, the dataset and even the hyper-parameters used. The authors chose several open-source LLMs and a few language tasks, but only partially addresses the concern. I would suggest the authors to extend their evaluation to other modalities such as images.
2. The paper only evaluates 3 parameter-efficient fine-tuning methods. I would suggest conducting a more thorough study to cover more parameter-efficient fine-tuning methods.

**Questions:**

N/A

---

> ### Author Response · Authors · 2023-11-14
> **Response to Reviewer JTKU**
>
> > I am mainly concerned that the results are not generalizable since these properties can heavily depends on the model architecture, the dataset and even the hyper-parameters used. The authors chose several open-source LLMs and a few language tasks, but only partially addresses the concern. I would suggest the authors to extend their evaluation to other modalities such as images.
>
> In our paper, we deliberately concentrate on exploring privacy and security concerns specific to Language Model Models (LLMs). This focused approach was chosen due to the significant computational demands already present at this limited scope. Expanding our investigation further would greatly intensify these demands. Nonetheless, we acknowledge the importance of similar issues in the image domain and are planning a dedicated study to address these in the near future.
>
> > The paper only evaluates 3 parameter-efficient fine-tuning methods. I would suggest conducting a more thorough study to cover more parameter-efficient fine-tuning methods.
>
> For our study, we have chosen to focus on the three most prevalent fine-tuning techniques, primarily due to the practical challenges involved in evaluating every possible method. However, we are open to suggestions from reviewers. If we missed a popular technique, we are willing to conduct additional experiments within a constrained setup to assess its efficacy and impact.

---

> ### Author Response · Authors · 2023-11-22
>
> We thank the reviewer again for their feedback and would like to ask if there is anything else that requires clarification. Additionally, we would like to reiterate that some of the fine-tuning techniques, such as ICL, do not have a counterpart in the image domain.

---

### Meta-Review · Area_Chair_pgMW · 2023-12-12

**Metareview:**

(a) Summarize the scientific claims and findings of the paper based on your own reading and characterizations from the reviewers.

This paper is a comprehensive approach to LLMs security evaluations. LoRA, Prompt tuning, and incontext learning are studied under the lens of privacy, security, and intellectual property. InContext Learning is vulnerable to membership inference attacks. Privacy defenses in incontext learning is important. All three training paradigms are vulnerable to model stealing attacks. The model can be easily stolen with GPT3.5-generated data. With respect to backdoor attacks, Low-Rank Adaptation (LoRA) and Soft Prompt Tuning (SPT) exhibit a higher susceptibility. These insights emphasize the necessity for tailored defenses in the deployment of LLM adaptation techniques.

(b) What are the strengths of the paper?

The paper systematically studies the security implications of various paradigms in finetuning LLMs. This is a topic of importance. The study is pretty comprehensive.

(c) What are the weaknesses of the paper? What might be missing in the submission?

The experiments are not meant to be novel. And that might be fine in some cases. What is troubling is that there is not much interpretation of the results. the paper stops at reporting the results. Given the lack of novelty, the evaluation needs to be thorough, which this paper fails at. The evaluations metrics are not thorough, and the settings are not state-of-the-art. Important evaluations are missing such as the accuracy of the resulting models.

**Justification For Why Not Higher Score:**

The paper is perhaps too ambitious and does not provide convincingly thorough empirical evaluations that it promises. The lack of interpretation of the findings makes it difficult to judge the contributions it is claiming to make.

**Justification For Why Not Lower Score:**

N/A

---

### Decision · Program_Chairs · 2024-01-16

Reject